# Sensing ecosystem dynamics via audio source separation: A case study of marine soundscapes off northeastern Taiwan

Tzu-Hao Lin[1]*, Tomonari Akamatsu[2], Yu Tsao[3]

1 Biodiversity Research Center, Academia Sinica, Taipei, Taiwan (R.O.C), 2 The Ocean Policy Research Institute, The Sasakawa Peace Foundation, Tokyo, Japan, 3 Research Center for Information Technology Innovation, Academia Sinica, Taipei, Taiwan (R.O.C)

* lintzuhao@gate.sinica.edu.tw

## Abstract

Remote acquisition of information on ecosystem dynamics is essential for conservation management, especially for the deep ocean. Soundscape offers unique opportunities to study the behavior of soniferous marine animals and their interactions with various noise-generating activities at a fine temporal resolution. However, the retrieval of soundscape information remains challenging owing to limitations in audio analysis techniques that are effective in the face of highly variable interfering sources. This study investigated the application of a seafloor acoustic observatory as a long-term platform for observing marine ecosystem dynamics through audio source separation. A source separation model based on the assumption of source-specific periodicity was used to factorize time-frequency representations of long-duration underwater recordings. With minimal supervision, the model learned to discriminate source-specific spectral features and prove to be effective in the separation of sounds made by cetaceans, soniferous fish, and abiotic sources from the deep-water soundscapes off northeastern Taiwan. Results revealed phenological differences among the sound sources and identified diurnal and seasonal interactions between cetaceans and soniferous fish. The application of clustering to source separation results generated a database featuring the diversity of soundscapes and revealed a compositional shift in clusters of cetacean vocalizations and fish choruses during diurnal and seasonal cycles. The source separation model enables the transformation of single-channel audio into multiple channels encoding the dynamics of biophony, geophony, and anthropophony, which are essential for characterizing the community of soniferous animals, quality of acoustic habitat, and their interactions. Our results demonstrated the application of source separation could facilitate acoustic diversity assessment, which is a crucial task in soundscape-based ecosystem monitoring. Future implementation of soundscape information retrieval in long-term marine observation networks will lead to the use of soundscapes as a new tool for conservation management in an increasingly noisy ocean.

**Data Availability Statement:** All recording clips used in this study are archived in the Geophysical Database Management System (http://gdms.cwb.gov.tw) and available for research use with

permission from the Central Weather Bureau of Taiwan. All of the long-term spectrograms and the source separation model that support the findings of this study are fully available without restriction on depositar (https://data.depositar.io/en/dataset/deep-water-soundscapes-off-northeastern-taiwan). MATLAB codes used in this study are available in Code Ocean (https://codeocean.com/capsule/7292152/).

**Funding:** THL was supported by the JSPS KAKENHI, Japan (Grant Number 18H06491) and the Ministry of Science and Technology, Taiwan (Grant Number MOST105-2321-B-001-069-MY3). The funders had no role in study design, data collection and analysis, decision to publish, or preparation of the manuscript.

**Competing interests:** The authors have declared that no competing interests exist.

## Author summary

Understanding the status of biodiversity and its changing patterns is crucial for conservation management. Ecological assessment in deep-sea environments is challenging owing to the limited accessibility and visibility. Nevertheless, we can listen to underwater sounds to investigate the acoustic behaviors of marine animals. This study applied machine learning techniques to analyze underwater recordings transmitted from a deep seafloor observatory off northeastern Taiwan. In particular, our computational approach does not require preexisting labels in the procedure of model building. The model can automatically learn acoustic features and effectively separate sounds of marine mammals, fishes, and shipping activities with minimal supervision. The analysis of 2.5 years of audio data revealed that the studied continental shelf environment has a high diversity of sound-producing animals, and the community structure changes with diurnal, lunar, and seasonal cycles. The applied techniques also generated a database of environmental and anthropogenic sounds, which can facilitate future investigations on how human activities affect the acoustic environment and biodiversity. Our technique is entirely based on passive acoustics, which minimizes interference with animals. It also transforms audio recordings into highly interpretable ecological data, which can be useful for tracking changes in marine biodiversity.

## Introduction

Understanding trends in marine ecosystems is essential for effective marine conservation [1]. Especially in the deep ocean, investigating marine biodiversity has been challenging owing to limited visibility and accessibility. Consequently, remote-sensing platforms that can acquire nonvisual data are crucial for efficient assessment of marine ecosystems. Hydroacoustic systems, including active and passive sensing, are widely used to reveal the dynamics of marine fauna and physical–biological coupling [2–4]. In particular, various fixed-point observatories have generated a vast volume of underwater recordings characterizing marine soundscapes [5,6].

Marine soundscapes, which are a composite of geophysical, biological, and anthropogenic sounds, have been used to assess marine biodiversity [7–11]. Long-term monitoring of marine soundscapes provides an opportunity to study the community of soniferous animals, including marine mammals, soniferous fishes, and crustaceans [12–14]. In the Southern Ocean, species-specific spectral peaks have been used to investigate the community structure and migration behavior of marine mammals over the seasons [15]. In shallow coastal waters, changes in sound pressure levels in low frequency range showed clear periodic patterns of fish choruses in diurnal and tidal cycles [16,17]. In subtidal reef environments, evaluation of the number of impulsive sounds revealed the temporally shifting patterns of the diurnal behavior of snapping shrimps [18]. These examples suggest that marine soundscapes contain ecological information that can be used to investigate the phenological patterns of soniferous animals and their responses to environmental changes.

Investigating sounds associated with geophysical events and human activities can also be used to evaluate the impacts of anthropogenic activities such as fishing and shipping on marine ecosystems [19–22]. The increase of environmental and anthropogenic sounds represents an indicator of habitat change, but it interferes with acoustic analysis by masking the received biological sounds [23]. To date, the data and metrics generated from soundscapes have not been effectively converted into tools that can be used by managers and stakeholders. One reason is

that conventional tools, including sound pressure levels and ecoacoustic indices, cannot prevent the mutual interference caused by geophony, biophony, and anthropophony, leading to difficulty in data interpretation [24,25].

To implement soundscapes in marine ecosystem assessments, the audio information relevant to the presence of soniferous animals must be separated from that of other noise-generating activities. In bioacoustics studies, researchers have applied a series of modeling approaches to develop detectors and noise filters. Successful applications rely on prior knowledge of detection targets and noise such as their dominant frequencies and signal durations for parameter tuning [26–28]. However, parameter tuning is labor intensive and often requires repetition when analyzing data collected from a new environment. Recent developments in deep learning technology have reduced the difficulty in classifying animal vocalizations and human speech [29–32], but most modern deep learning techniques require numerous preexisting labels for supervised training. Moreover, supervised learning cannot identify sound sources or anomalies that are excluded in the training dataset. While soundscapes are understudied in many marine environments, supervised learning will remain of limited application until a comprehensive labeled database is established.

Unsupervised learning techniques represent another potential solution for conducting soundscape-based ecosystem assessment. Clustering is a machine learning technique used to identify groups of similar objects in a set of unlabeled data. It has been shown to improve the content interpretation of environmental recordings [33] and reveal the species identity of dolphin echolocations [34]. However, clustering performance is subject to the selection of acoustic features and the recording of simultaneous sound sources [35]. Two signals recorded within the same time frame may distort the observed acoustic features and lead to inappropriate inductive inference. The issue of simultaneous source interference may be addressed by applying blind source separation (BSS). A BSS model aims to separate a set of sound sources from a mixture, without prior information specific to individual sources or the mixing process [23]. Unlike clustering, effective BSS can reconstruct individual sound sources and add value to noisy recordings. Robust principal component analysis (RPCA) has been successfully applied to separate accompaniment and singing voice by assuming the accompaniment to be in a low-rank subspace and the singing voice to be relatively sparse [36]. When such low-rank assumption does not hold (e.g., when biotic and abiotic sounds are highly dynamic), the RPCA cannot yield satisfactory performance. Independent component analysis (ICA) is another widely used BSS model that can separate statistically independent components [37]. ICA works well when provided with multichannel audio, but most fixed-point acoustic observatories only have one hydrophone at each location.

BSS in monaural soundscape recordings can be achieved by applying nonnegative matrix factorization (NMF) to model source-specific timbre and temporal patterns [38,39]. Lin et al. [40] developed the periodicity-coded NMF (PC-NMF) to learn a set of spectral features essential to the time–frequency representations of long-duration recordings and to identify subsets of spectral features that display unique periodicity patterns. With the assumption of source-specific periodicity, biological chorus and noise can be effectively separated in an unsupervised manner [11]. The PC-NMF can also separate sound sources with varied patterns of periodical occurrence, revealing a more delicate temporal pattern of acoustic diversity [23].

This study investigated the application of using soundscapes to assess the temporal dynamics of biological and anthropogenic activities in a continental shelf environment. We addressed the challenges in analyzing long-duration underwater recordings, including simultaneous source interference and the lack of a comprehensive labeled database. First, we review the current knowledge on marine soundscapes off northeastern (NE) Taiwan. Second, we detail the application of audio visualization, source separation, and clustering techniques to assess the

diversity of biotic and abiotic sounds. Finally, we discuss the advancements of soundscape-based ecosystem assessment that involves leveraging the techniques of audio source separation.

## Marine soundscapes off northeastern Taiwan

The marine ecosystem off NE Taiwan hosts a rich diversity of marine life owing to complex bathymetry and the influence of the Kuroshio Current. For example, sighting and stranding reports from 1994 to 2012 revealed at least 21 species of marine mammals in NE Taiwan waters [41]. Studies on cetacean vocalizations were based on audio data recorded by the Marine Cable Hosted Observatory (MACHO), a long-term marine observatory at the Ilan Ridge (Fig 1). Echolocation clicks and whistles of odontocetes were frequently detected in frequencies higher than 20 kHz [42] and within the range of 4.5–48 kHz, respectively [43]. Whistle usage displayed significant diurnal and seasonal variations, and the phenological patterns were associated with changes in the group behavior and species composition of delphinids [43,44]. In addition to the sounds of marine mammals, the MACHO repeatedly recorded one type of fish chorus between 2 and 3 kHz after sunset [45]. However, the observation was based on 1-month recordings, and thus, seasonal changes could not be identified.

Sounds of geophysical events are essential components of marine soundscapes off NE Taiwan. Earthquake-induced underwater sounds are evident in the infrasound range because of active interactions between the Philippine Sea Plate and the Eurasian Plate [46]. The climate of NE Taiwan is heavily influenced by the northeastern monsoon, especially from fall to winter. The northeastern monsoon brings strong winds and heavy rainfall, significantly affecting the range between 0.5 and 5 kHz [42]. Shipping activity is another vital source of abiotic sounds. Suao and Nanfangao Ports are the largest industrial and fishing ports in Ilan County. From 2011 to 2014, approximately 1200 containers were transported to and from Suao Port each year, and more than 700 fishing vessels were registered at Nanfangao Port.

The MACHO recordings represent a valuable database for investigating soundscapes off NE Taiwan. However, the entire dataset has yet to be extensively investigated, and acoustic analysis is challenging owing to the continuous interference of highly variable electrical noise [47]. Moreover, the recordings contain multiple types of biotic and abiotic sounds that may be overlapping in time or frequency domains, and each sound source also possesses spectral variability. In order to reduce the interference among sound sources, the present study applied audio source separation in the information retrieval of soundscapes and investigated the patterns of acoustic phenology and the diversity of cetacean vocalizations, fish choruses, and abiotic sounds across 2.5 years of MACHO recordings.

## Results

From October 9, 2011, to May 15, 2014, the MACHO collected 946 days of underwater recordings. We generated a long-term spectral average (LTSA) based on 30-s and 93.75-Hz resolution. Nevertheless, the visualization of the entire dataset still requires at least 36 GB of memory. To reduce computational resource burden, we selected 12 days of audio data to generate an LTSA. Subsequently, we applied the PC-NMF to learn 90 spectral features from the LTSA and then separated them into four independent sound sources.

### Source separation performance

According to their spectral characteristics, the four sound sources were abiotic sounds, fish choruses, cetacean vocalizations, and electrical noise generated by the MACHO (Fig 2). The source separation model correctly identified 67.33% of cetacean vocalizations when the false-

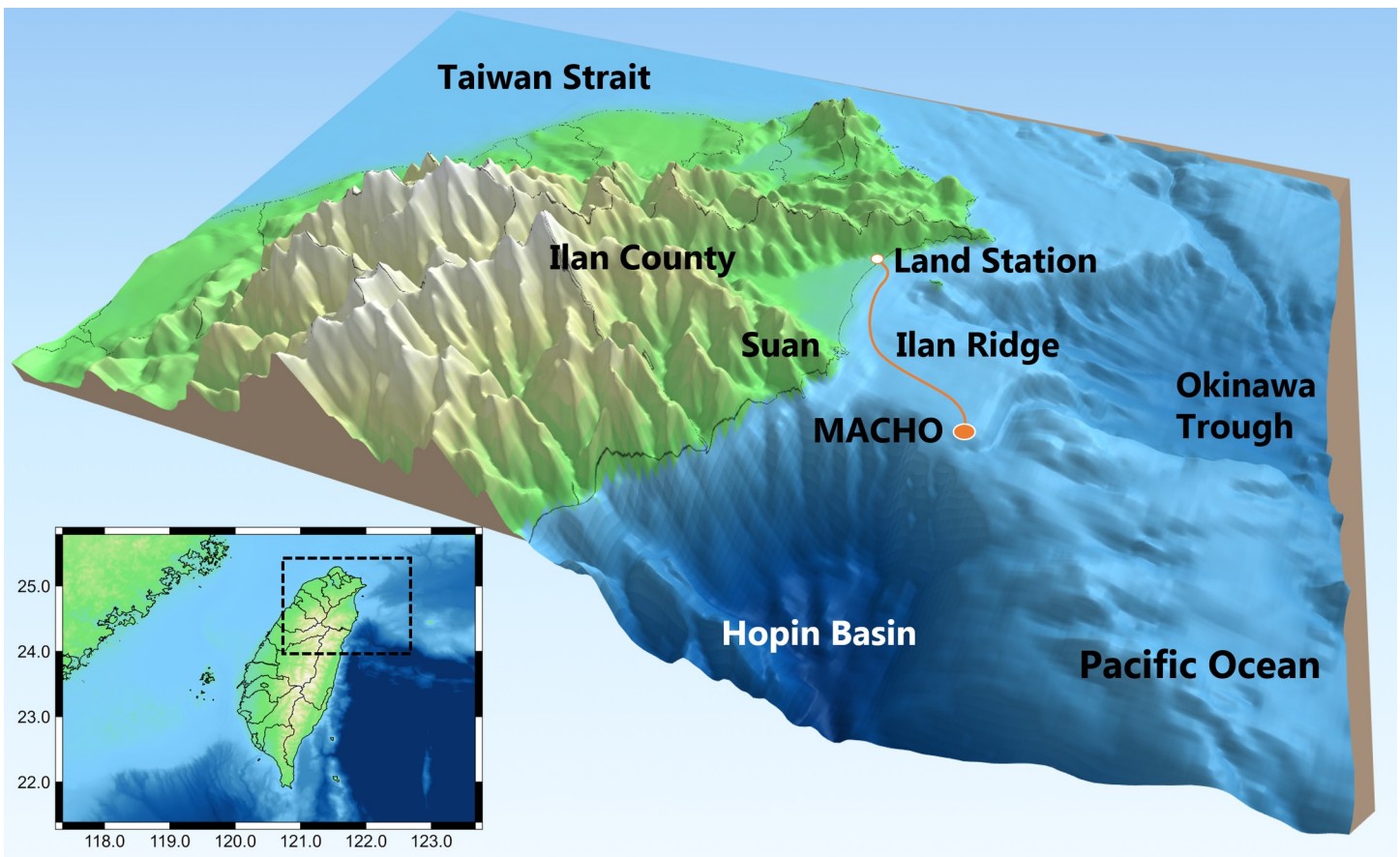

**Fig 1. Marine Cable Hosted Observatory.** The map shows the location of the MACHO and bathymetry in NE Taiwan waters. From 2011 to 2014, a bottom-mounted hydrophone was located approximately 28 km east of Ilan County. Underwater recordings were transmitted to a land station through a 45-km-long submarine cable.

positive rate was set to 5%, but it failed to separate fish choruses and low-frequency abiotic sounds. We investigated the temporal activations and manually adjusted the source indicators of the two spectral features associated with fish choruses. After the adjustment, the model out-performed conventionally applied band-pass filters (S1 Fig) and correctly identified 73.10% of fish choruses and 85.99% of cetacean vocalizations at the same false-positive rate (5%).

Analysis of the entire dataset revealed that the four sound sources were overlapping in the frequency domain but still displayed different spectral features (Fig 3). Abiotic sounds mainly extended from low frequencies to 20–30 kHz. Fish choruses were recorded in the frequency range from 500 Hz to 3 kHz, with two peaks at 750 Hz and 2.5 kHz. Cetacean vocalizations were primarily detected at $\geq$ 4 kHz, with prominent spectral variations in the ultrasonic range. Electrical noise consisted of spectral peaks distributed in various frequencies and displayed a highly unpredictable temporally changing pattern. A few noise-related spectral peaks were more predictable in a day, but their frequencies also changed between months.

## Acoustic phenology and temporal correlations

The relative intensities of abiotic sounds, cetacean vocalizations, and fish choruses during the 2.5 years are presented in Fig 4. The audio data from March 24 to July 1, 2013, recorded high-intensity broadband noise in frequencies $\geq$ 30 kHz, leading to incorrect detection of cetacean vocalizations and adding noise in the analysis of time-lagged correlations. Because the high-

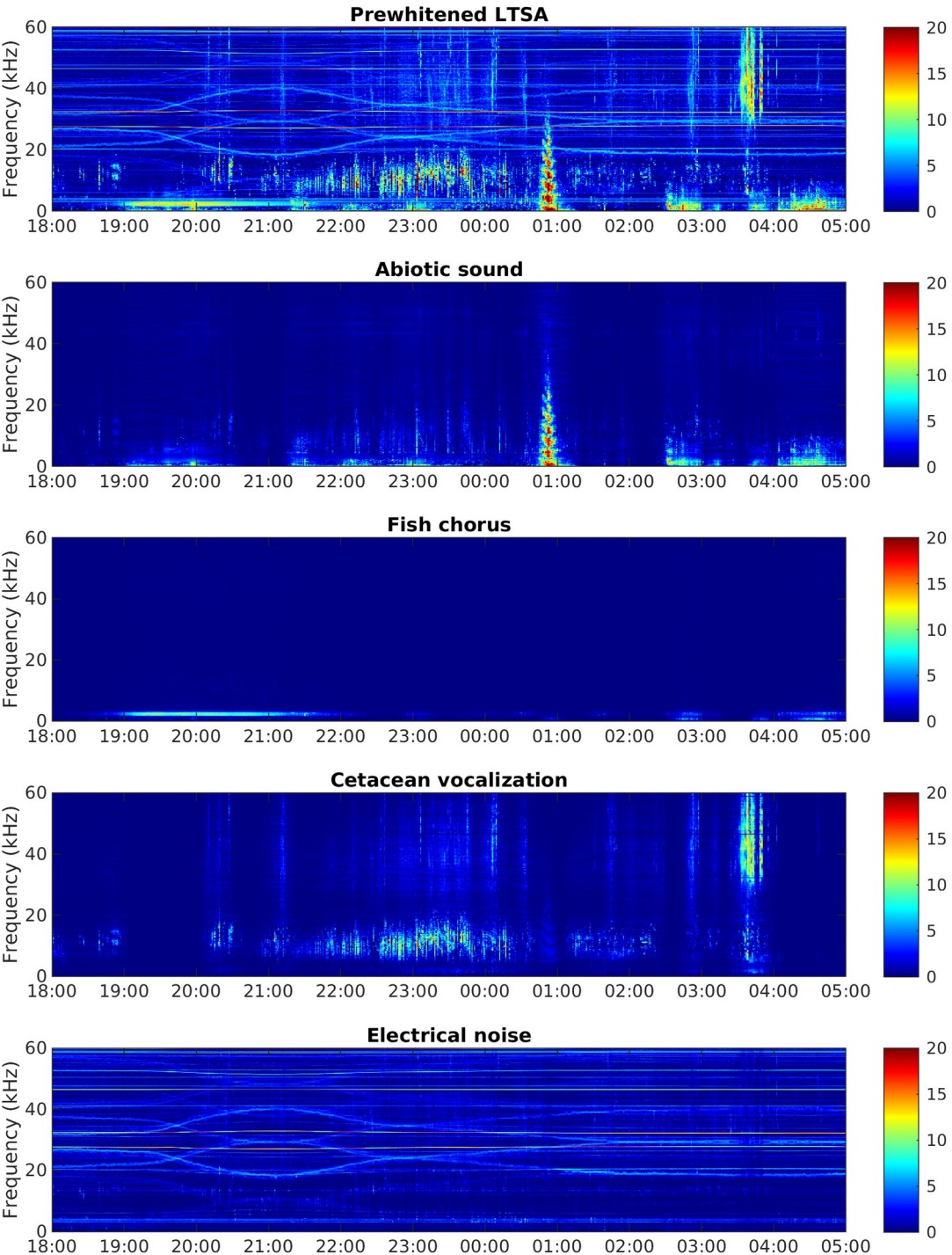

**Fig 2. Example of audio source separation.** The x- and y-axes denote the time and frequency, respectively, and the color denotes the relative intensity of the acoustic signal in the time–frequency plane. The uppermost panel represents a prewhitened LTSA from June 21 to 22, 2012, and the other panels represent the four sound sources separated from the prewhitened LTSA.

frequency range was too heavily corrupted to identify echolocation clicks manually, the intensities of cetacean vocalizations were manually set to 0 to prevent bias in the phenological pattern analysis. The time-lagged correlations indicated that abiotic sounds, cetacean

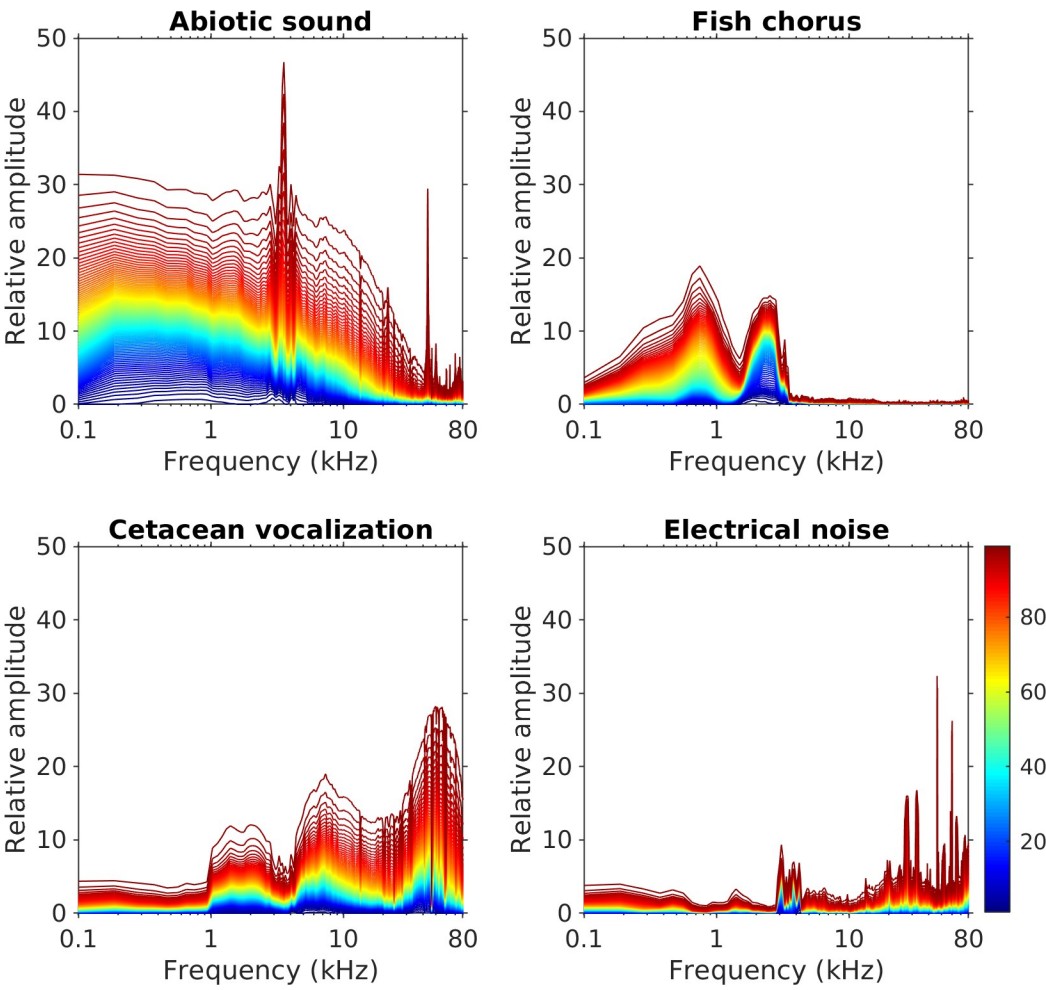

**Fig 3. Spectral variations of abiotic sounds, fish choruses, cetacean vocalizations, and electrical noise.** Different colored lines represent the cumulative probability distribution of spectral amplitudes from the 1st to the 99th percentiles. The relative amplitudes, which represent the signal strength displayed on the prewhitened LTSA after performing source separation, were measured from the entire dataset of MACHO recordings.

vocalizations, and fish choruses displayed different periodic patterns (Fig 5). Compared with the nonperiodical pattern observed in abiotic sounds, the two biophonic sound sources exhibited clear periodic structures. The fish chorus had recurring peaks at 1-day, 14-day, 27-day, and 1-year intervals. The recurring pattern of cetacean vocalizations was slightly different, with peak values at 1-day, 28-day, and 1-year intervals.

Analyses of the time-lagged correlations for the three pairs of sound sources suggested that significant temporal correlations were not identifiable between abiotic sounds and fish choruses, or between abiotic sounds and cetacean vocalizations. However, peak correlation values at intervals of 0.17 and 186 days were noted between fish choruses and cetacean vocalizations. The intensity of fish choruses was highest at 8 PM, whereas that of cetacean vocalizations was highest at 1 AM (Fig 6). The difference in diurnal behaviors resulted in the 0.17-day peak, as observed in the time-lagged correlations. In addition, the differences in the seasonal changes in intensities between fish choruses (highest at week 28) and cetacean vocalizations (highest at week 6) reflected the 186-day peak in the time-lagged correlations.

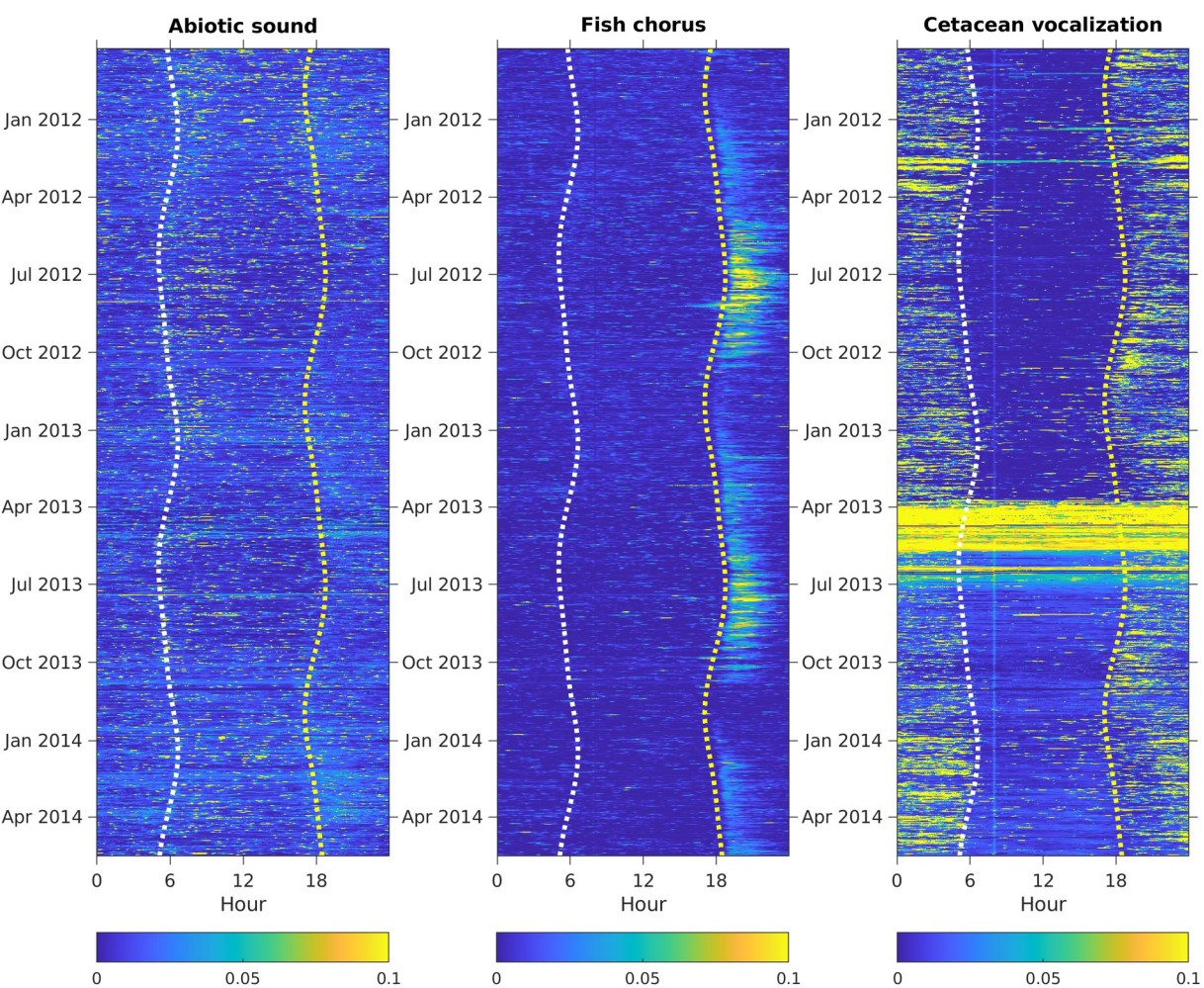

**Fig 4. Temporal changes in abiotic sounds, fish choruses, and cetacean vocalizations.** The relative amplitudes were normalized to a range between 0 and 1. Seasonal changes in sunrise and sunset are shown as white and yellow dashed lines, respectively.

### Diversity of abiotic and biotic sounds

By applying a threshold of 0.05 to the relative intensities, we could identify 9211, 1490, and 6410 events for abiotic sounds, fish choruses, and cetacean vocalizations, respectively. The applied threshold ensured that the false alarm rates of the detected fish choruses and cetacean vocalizations were $\leq$ 5%. On the basis of the mean spectrum of the extracted events, we analyzed the spectral diversity of each sound source through $k$-means clustering. Abiotic sounds were separated into five clusters, which explained 95.17% of the variation (Fig 7). Among them, the first cluster was characterized by low-frequency sounds ($<$ 2 kHz) and was recorded more frequently after July 2013. The second to fourth clusters were associated with the underwater radiated noise of moving vessels, and many events recorded 43.4 or 50 kHz narrowband sounder signals. The last cluster was characterized by narrowband sounder signals (peak frequency at 3.5 kHz or 43.4 kHz) and was mainly recorded from 5 AM to 10 AM, and from May to October.

We identified three clusters of fish chorus ($R_{k = 3}$ = 93.28%) and six clusters of cetacean vocalizations ($R_{k = 6}$ = 95.07%) with unique spectral and temporal patterns. According to our manual inspection, the third fish chorus cluster was a mixture of the other two clusters. The

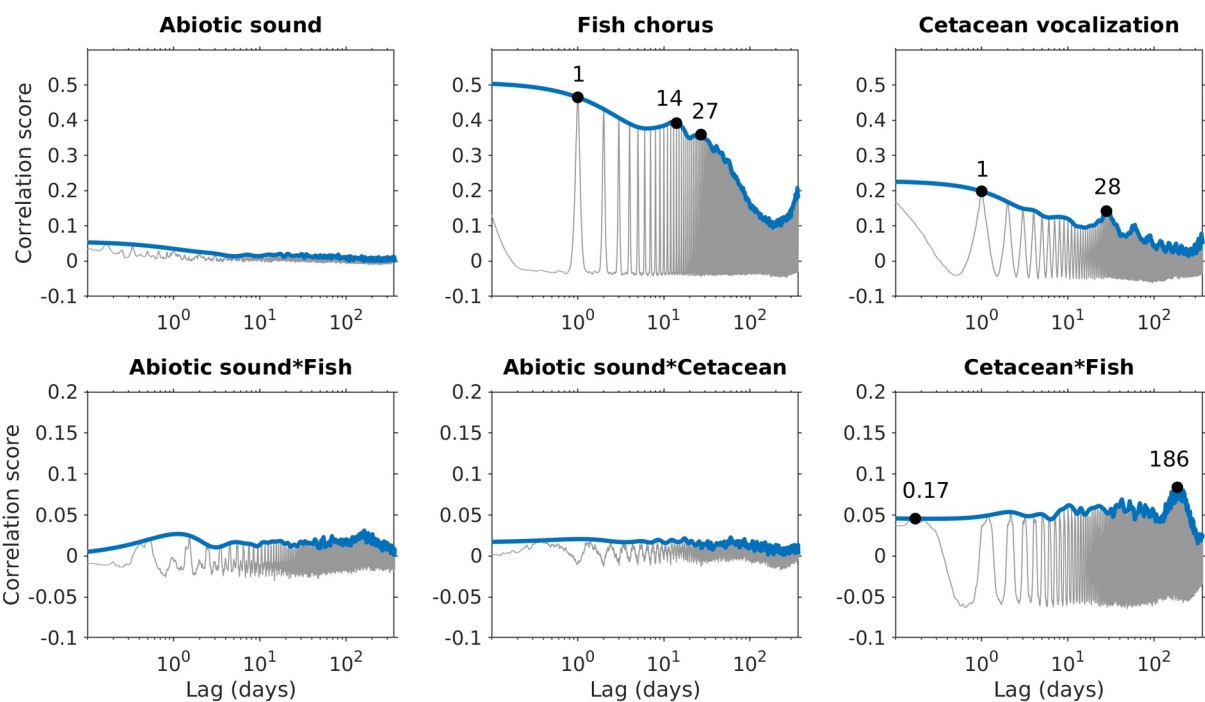

**Fig 5. Source-specific periodic patterns and temporal correlations between pairs of sound sources.** Gray lines represent the time-lagged correlation scores computed using autocorrelation (intra-source) or cross-correlation (inter-source). Blue lines represent the envelope of the time-lagged correlation scores using a 1-day sliding window.

seasonal distribution revealed that this cluster was primarily detected in the transitional phase (August) between the main occurring periods of the second cluster and the first cluster. However, the first cluster during daytime was falsely detected because of the interference of shipping noise.

The clustering results of cetacean vocalizations indicated a high diversity of echolocation clicks. The first four clusters were dominated by high-frequency echolocation clicks (i.e., > 25 kHz), displaying different patterns of peak frequency and frequency bandwidth. In addition, low-frequency tonal sounds (i.e., 4–20 kHz) were observed in these clusters, suggesting that they were produced by delphinids. The fifth cluster was characterized by median-frequency echolocation clicks (i.e., 10–50 kHz). The sixth cluster was dominated by low-frequency echolocation clicks or tonal sounds (i.e., 4–18 kHz). Furthermore, the diurnal and seasonal variations revealed higher acoustic diversity during the night and from July to April.

## Discussion

The underwater recordings collected by MACHO, comprising various cetacean vocalizations, fish choruses, and abiotic sounds, represent the soundscape dynamics in the continental shelf environment off NE Taiwan. Our source separation model revealed evident time-lagged correlations between soniferous fish and cetaceans owing to the difference in phenological patterns. The source separation results also allowed for a more comprehensive assessment of acoustic diversity. The applied technique of audio source separation could generate broad and major implications by transforming single-channel audio into multiple channels that encode the dynamics of biophony, geophony, and anthropophony. The source-specific information will provide new opportunities for using ecoacoustics to investigate biodiversity and ecological interactions.

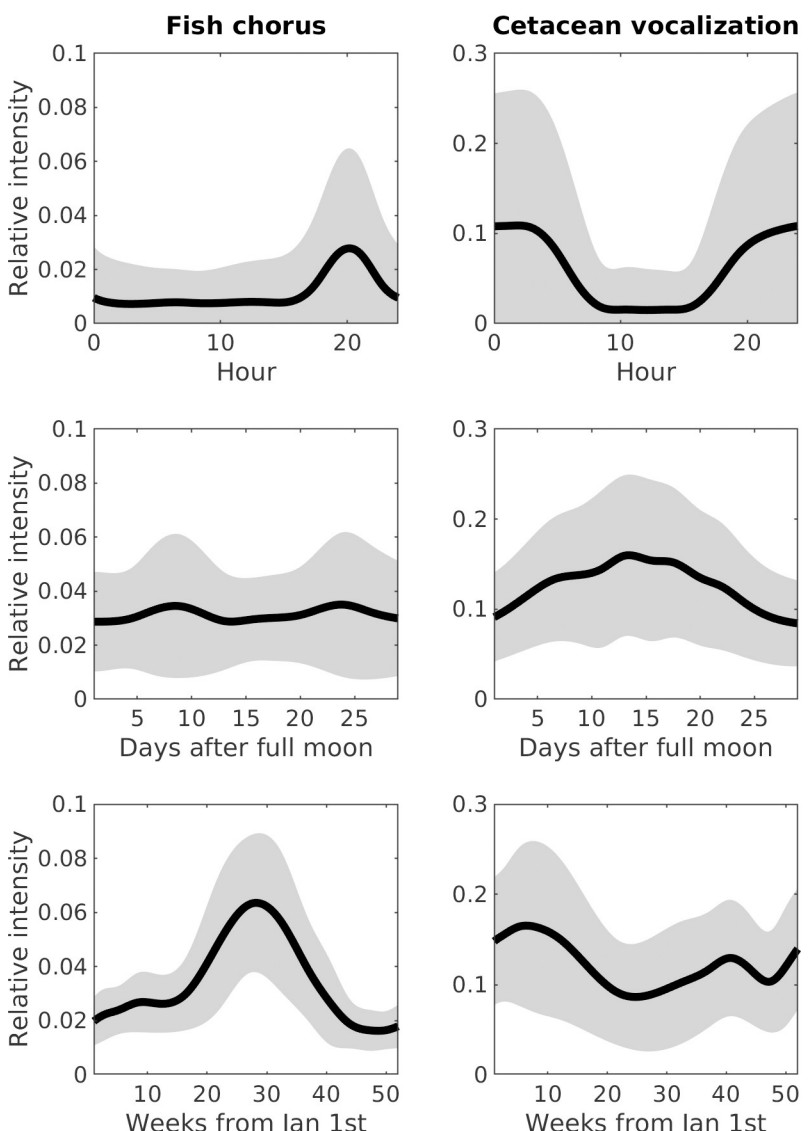

**Fig 6. Acoustic phenology of cetacean vocalizations and fish choruses.** Panels show the modeling of relative intensities in diurnal, lunar, and seasonal cycles. Thick lines represent the mean intensities, and gray areas represent the range of standard deviations.

## Applying audio source separation to assess soundscapes

The potential applications of soundscapes are enormous because of the various sound-producing activities recorded. However, automatic identification of sound sources remains a crucial step. Most audio classification techniques are trained to identify categorical labels, and their performance may decline when multiple sound sources are recorded [48]. Audio source separation aims to reconstruct independent sound sources from a mixture. Therefore, the separation result not only reduces the interference among sound sources but also facilitates further applications, such as variability assessment of source behaviors [23]. State-of-the-art speech separation models based on deep learning were trained to map noisy recordings to clean speech. However, a database of clean sounds is unlikely to be obtained in marine environments [49].

**Abiotic sound**

**Fish chorus**

**Cetacean vocalization**

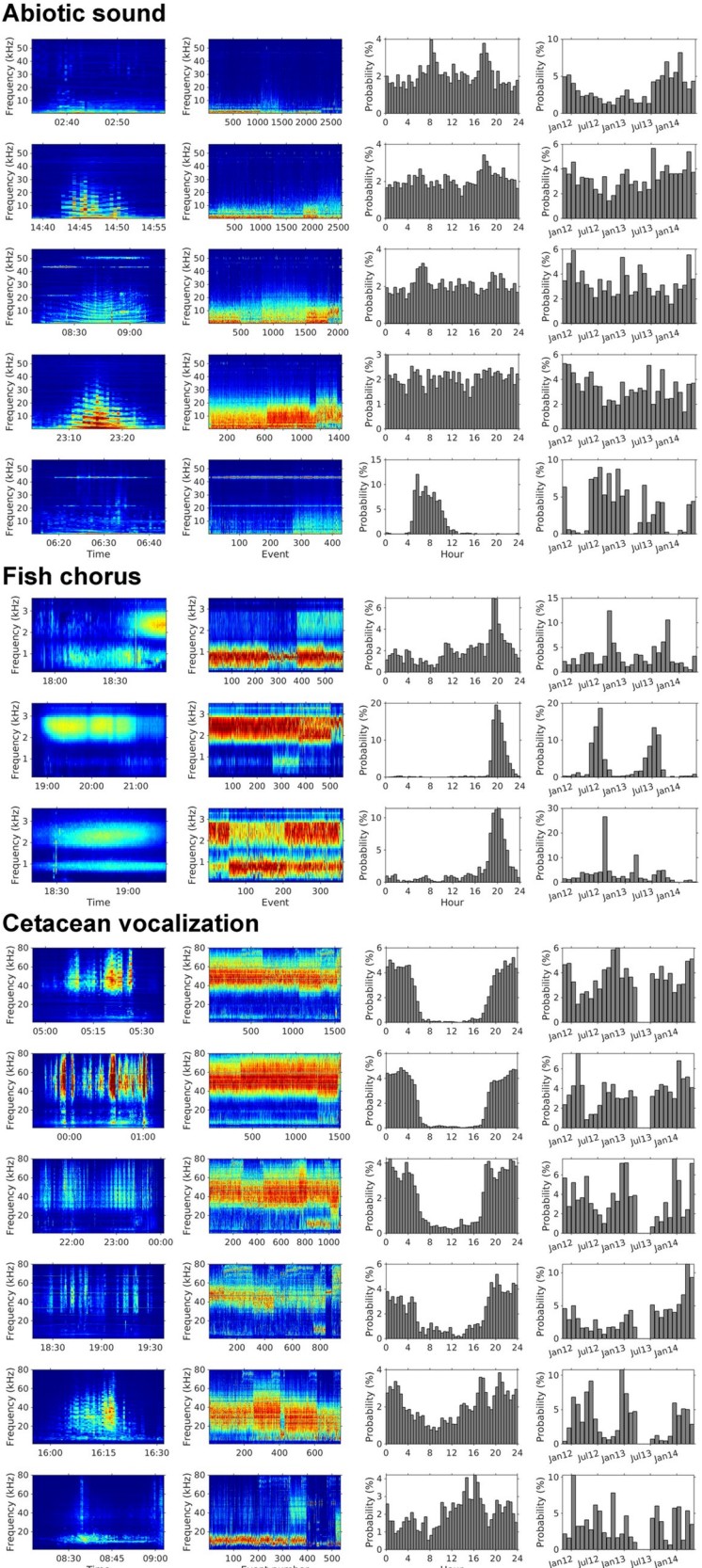

**Fig 7. Clustering of abiotic sounds, fish chorus, and cetacean vocalizations.** Rows represent the audio clusters identified from the MACHO recordings. The first column displays the examples of LTSA reconstructed using our source separation model. The second column exhibits the mean spectra of acoustic events associated with each cluster. The third column presents the changes in detection probabilities in 24 h. The fourth column illustrates the changes in detection probabilities in each month. The probability was calculated by dividing the detected duration in each time category by the total duration of each audio cluster.

This study demonstrated that the assumption of source-specific periodicity is effective for the BSS of marine soundscapes because the PC-NMF can differentiate the different periodic patterns between biotic and abiotic sounds. Changing the source indicators in a semi-supervised manner further improved the model to recognize the varied diurnal and seasonal patterns of marine mammal vocalizations and fish choruses. Therefore, the applied techniques should work for acoustic data that display apparent diversity in phenological patterns. Although the recurring behavior of wildlife varies based on various biotic and abiotic factors, the PC-NMF may still help us examine the evolution of acoustic phenology in response to a changing environment.

The performance of PC-NMF depends on the unsupervised learning of spectral features. Feature learning is constrained by the choice of sparsity and the number of spectral features. A lower sparsity forces the NMF representation toward a more global structure than a parts-based representation [50]. In the case of soundscapes, tonal signals are sparser than broadband signals in the frequency domain. Our experiences with marine and terrestrial soundscapes suggest that a sparsity between 0.3 and 0.5 generally produces a dictionary that well reflects acoustic diversity. Future applications can choose a sparsity if prior knowledge is available. The number of spectral features applied in this study is an arbitrary decision. A simulation showed that PC-NMF performed relatively worse when fewer than 60 spectral features were used [40]. Therefore, we expected a larger dictionary to retain the acoustic diversity of real-world data.

The dictionary of spectral features learned by the first NMF layer fits our prior knowledge of the four sound sources (Fig 8). The dictionary also suggests that only two spectral features are required to reconstruct fish choruses. The unsupervised learning results corresponded to our observation that the third cluster is a mixture of the other two clusters. The results demonstrated the capability of PC-NMF to learn the general pattern of acoustic diversity. However, PC-NMF is incapable of learning every audio signal because the learning objective is to minimize reconstruction errors. Therefore, it tends to ignore rarely occurring sounds in long-duration recordings. Despite this, the integration of audio source separation and clustering still provides an efficient solution to build an annotated acoustic library from long-duration recordings.

Another advantage of PC-NMF is the visualization of model parameters, particularly spectral features and their associated temporal activations, which elevates the model interpretability and enables experienced users to improve the performance of unsupervised learning. As demonstrated in this study, the source separation performance improved when the source indicators learned by the second NMF layer were manually corrected. This can be done by inspecting the learned temporal activations of the first NMF layer to determine whether a source indicator is appropriate. In some cases, manually adjusting model parameters are necessary because our assumption of source-specific periodicity may not be able to solve a complicated dataset. The flexibility to develop a model in a semi-supervised manner allows users to combine models trained from different datasets and manipulate their source indicators to improve the generalizability of audio source separation.

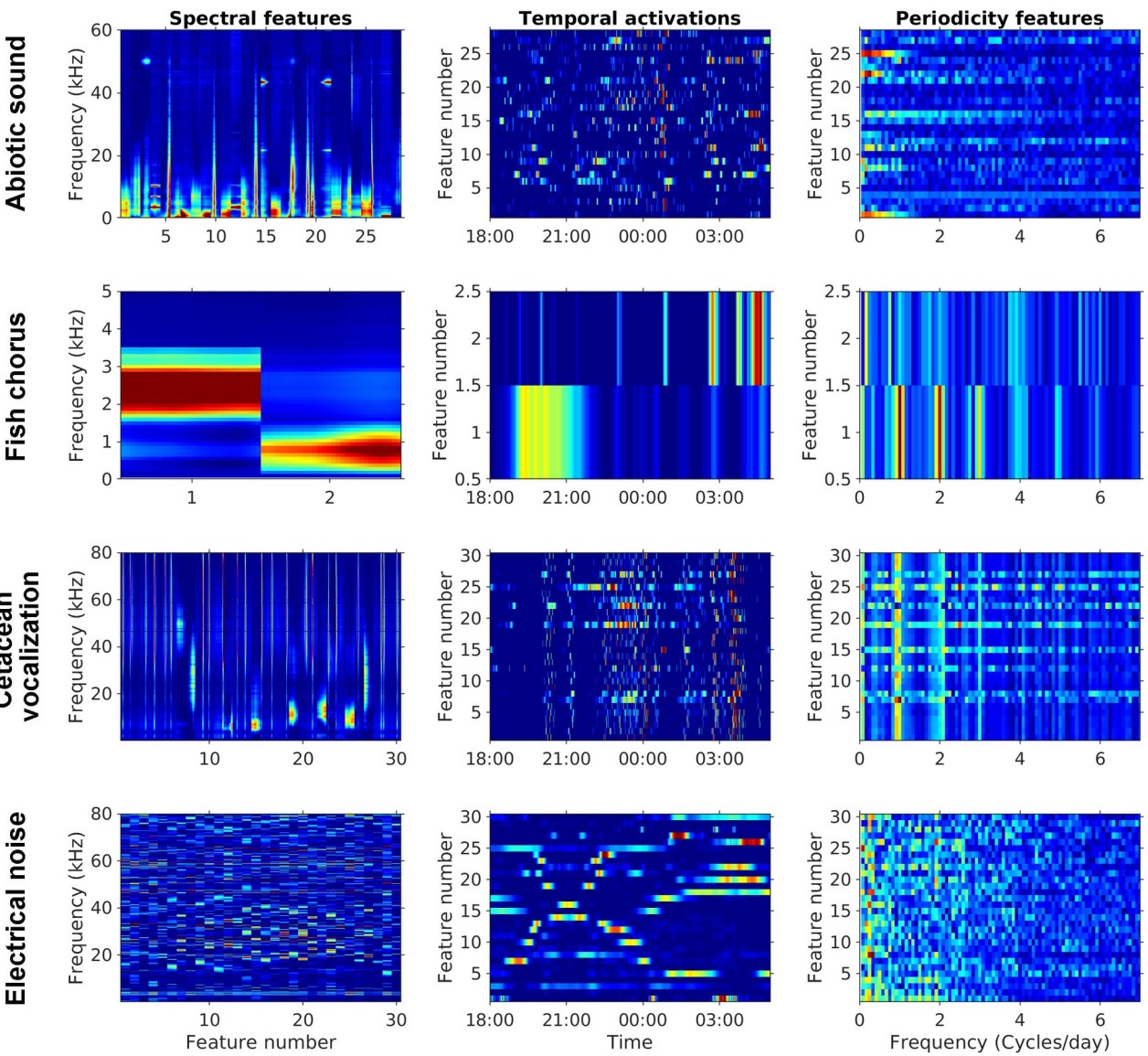

**Fig 8. Visualization of the parameters learned in the PC-NMF.** Panels in the first column represent the learned spectral features associated with abiotic sounds, fish choruses, cetacean vocalizations, and electrical noise. The second column represents the learned temporal activations associated with the prewhitened LTSA illustrated in Fig 2. The third column represents the learned periodicity features obtained from 12 days of training data.

## Retrieving ecological information from soundscapes

One of the challenges in soundscape-based ecosystem monitoring is the lack of ground truth data on soniferous animals. Without identifying the targets and their acoustic features, passive acoustics has limited application in the appraisal of biodiversity. Using soniferous fish as an example, one study surveyed fish sounds in the coastal waters of Taiwan but found none fish sounds off eastern Taiwan [51]. If we applied conventional bioacoustics approaches (e.g., a template detector), there would be no template to search for the fish chorus from 2.5 years of audio data. The techniques applied in this study allowed us to identify two types of fish chorus and determined their spectral, diurnal, and seasonal patterns with minimal prior knowledge. The second type of fish chorus, which has a peak frequency of 2.5 kHz, displays similar spectral

and diurnal patterns to the chorus of Myctophidae fishes recorded at Perth Canyon [52]. While the species of soniferous fish is still waiting for investigations, the acoustic data can assist future studies by selecting a period with a high detection rate.

Obtaining information relevant to the community of marine animals and its changing patterns is one of the goals of marine ecosystem monitoring. The clusters of cetacean vocalizations identified in this study can be considered a proxy for cetacean diversity in NE Taiwan waters. These clusters may represent clicks and whistles produced by delphinids that are commonly sighted in this region, including spinner dolphins (*Stenella longirostris*), spotted dolphins (*Stenella attenuata*), common dolphins (*Delphinus* spp.), Fraser's dolphins (*Lagenodelphis hosei*), Risso's dolphins (*Grampus griseus*), false killer whales (*Pseudorca crassidens*), and bottlenose dolphins (*Tursiops truncatus*) [41,44]. In addition, we identified clicks of sperm whales (*Physeter macrocephalus*) from the fifth and sixth clusters. Despite this, identifying the species responsible for each cluster remains challenging because a database of echolocation clicks in NE Taiwan waters is unavailable.

Even without knowing the species identity, the observed acoustic phenology can help us evaluate the changes in the community structure and behavior over time. The consistent nighttime foraging behavior and seasonal variation of sound types suggest that the studied continental shelf environment represents an essential habitat for many cetaceans, including resident and migrating species. Although similar seasonal and diurnal patterns have already been reported using a tonal sound detector and 1-year MACHO recordings [43,44], one detector requires extensive fine-tuning of the detection parameters and the removal of false alarms. In contrast, our audio source separation and clustering techniques considerably reduce the labor-intensive nature of the works. Furthermore, the data generated improve our understanding on the structure and variability of biotic sounds, which allow future studies to efficiently investigate the behavior of target species.

Continuous long-term recordings provide unique opportunities to investigate ecological interactions on various temporal scales. Our results indicate that the behaviors of cetaceans and soniferous fish are driven by a combination of different periodicities, from diurnal and lunar to seasonal cycles. Periodic patterns, such as the timing of occurrence, recurring interval, and recurring consistency, are the key parameters for predicting the status and trends of wildlife [53,54]. Diurnal and seasonal interactions were clearly observed between the acoustic presence of cetaceans and soniferous fish. While delphinids can acquire acoustic information regarding the identity, abundance, and location of their prey through passive listening [55,56], soniferous fish may also suppress their calling behavior when they detect the sounds of their predators [57]. Therefore, the time-lagged cross-correlations observed between cetacean vocalizations and fish choruses may be caused by the predator–prey interactions. To date, we have only analyzed the temporal correlations based on three sound sources. Further investigations of the temporal correlations between clusters may reveal more noteworthy ecological interactions and behavioral responses, such as competition for habitat or resources, to environmental change.

Monitoring environmental changes due to anthropogenic impact is crucial for marine conservation. In particular, for high-sea areas or deep-sea environments, where information on marine biodiversity and anthropogenic impact is minimal, the monitoring of marine soundscapes can generate high-resolution time series for assessing interactions between soniferous animals and anthropogenic activities [22]. This study demonstrated that the continental shelf environment off NE Taiwan was heavily influenced by sounds generated from shipping and fishing activities. Frequently occurring anthropogenic noise can interfere with the communication, prey detection, predator avoidance, reproduction, and habitat selection of marine organisms [58,59]. As ocean noise continues to increase, noise-induced physiological and

behavioral impacts may be observed more frequently. Information specific to anthropogenic noise will be a key indicator for predicting the ecological impacts due to anthropogenic activities.

## Future perspectives

In this study, we integrated the visualization of long-duration recordings, audio source separation, and clustering techniques to assess the soundscape dynamics of a continental shelf environment. Despite having data from only one seafloor observatory, the technique of audio source separation can be applied to ecoacoustics or bioacoustics research that has to contend with interference among sound sources. By transforming underwater recordings into channels specific to biotic and abiotic sounds, the output and associated indices are easier to understand for managers and stakeholders, who may not be familiar with acoustics. In addition, the ability to reconstruct spectral–temporal representations of individual sound sources enables the subsequent identification of various events. For example, the clustering of abiotic sounds revealed narrowband sounder signals, likely representing the operations of fish finders and sub-bottom profilers. Information specific to the behavior of fishing vessels can provide insights into the strategy of local fisheries, which is vital for effective marine spatial planning [60]. Therefore, the applied techniques of soundscape information retrieval can facilitate the analysis of acoustic data generated from the global network of seafloor observatories and improve marine affair-related decision making.

Numerous acoustic data are generated from various habitats and ecosystems. Advanced machine learning models are required to separate sound sources that are specific to different habitats and ecosystems. The recent development of deep learning techniques has significantly improved the retrieval of semantic information from audio and visual datasets [61]. To reduce dependency on numerous preexisting labels, weakly supervised or semi-supervised learning techniques may be considered in the future development of soundscape information retrieval [62]. Another feasible solution is to introduce domain knowledge into unsupervised learning or self-supervised learning [23]. Although this study introduced a periodicity constraint into audio source separation, integrating other ecological observations or hypotheses, such as the acoustical niche hypothesis, with deep learning may yield further developments in soundscapes to improve their efficacy as a conservation tool for biodiversity.

Scalability represents another issue in the analysis of a large amount of acoustic data. In this study, we used an NVIDIA QUADRO RTX 6000 GPU with 24 GB of VRAM to develop the source separation model. The GPU finished building the model in 13.2 min, but the same computation task took 27.6 min when using only a CPU (Intel Xeon W-2133 3.6GHz). A GPU with massive VRAM allows us to use more data to train an NMF-based source separation model and increase the amount of time-dependent information learned by each spectral feature. Theoretically, this should improve the generalizability of the model to various soundscape conditions. If such a GPU is not available, then the size of the training data for model building would have to be scaled down, and the resulting model may not be robust enough to handle changing soundscapes. However, a smaller model that can provide comparable performance is crucial for real-time or near real-time applications. Therefore, we integrated the codes used in this study into the Soundscape Viewer as an open-source toolbox to initiate the collaborative development of soundscape information retrieval and facilitate soundscape analysis in future studies.

## Materials and methods

### Marine cable hosted observatory

The MACHO is a long-term cabled observatory system used to monitor earthquakes and tsunamis off NE Taiwan [63]. In addition to seismometers, a bottom-mounted hydrophone

(model TC-4032; Reson, Slangerup, Denmark) was placed at the environmental sensing node, located at a depth of 277 m (24˚33.0′ N, 122˚07.9′ E). The environmental sensing node was in operation from October 2011 to May 2014. The effective sensitivity of the hydrophone was −164 dB re 1 V/μPa, and the frequency response ranged from 10 Hz to 80 kHz (±2.5 dB). Underwater recordings were sampled at 384 kHz and saved in a waveform audio format every 30 s.

## Visualization of long-duration recordings

To investigate long-duration recordings, we used the LTSA to visualize the spectral–temporal variations. First, we generated a magnitude spectrogram $S(f,t)$ (time resolution = 10.67 ms, frequency resolution = 93.75 Hz) for each 30-s audio clip using the discrete Fourier transform (DFT: Hamming window, FFT size = 4096 samples, overlap = 0). The magnitude spectrogram was then averaged over the entire 30-s period and transformed into a log-scaled mean spectrum $P(f)$ to characterize the spectral variation:

$$P(f) = 10 \times \log(mean(S(f,t))), \tag{1}$$

where $t$ and $f$ represent time and frequency, respectively in each 30-s period. All log-scaled mean spectra were combined consecutively in the LTSA [64,65]. Owing to the frequency sensitivity of the applied hydrophone, we only analyzed the frequency range $<$ 80 kHz.

Subsequently, we estimated the background noise by measuring the $n_{10}(f)$: the 10th percentile at each frequency bin $f$. All log-scaled mean spectra $P(f,t)$ were subsequently prewhitened by subtracting the noise floor $n_{10}(f)$ at each frequency bin, with negative values converted to 0:

$$\check{P}(f,t) = P(f,t) - n_{10}(f). \tag{2}$$

The removal of the background noise thus equalized the spectral gradient due to low-frequency ambient noise and reduced electrical noise with fixed-frequency peaks.

## Audio source separation

To reduce the interference among sound sources, we applied PC-NMF to analyze the prewhitened LTSA. PC-NMF is a tool for BSS based on two layers of NMF [40]. The first layer, representing a feature learning layer, can learn to decompose a nonnegative spectrogram into a set of spectral features ($W$) and associated temporal activations ($H$) by iteratively minimizing the reconstruction error. The temporal activations of each spectral feature were subsequently converted to periodicity domains using DFT. The matrix of periodicity was used as the input for the second layer. The second layer, which functions as a source-recognition layer, can learn to decompose the transpose matrix of periodicity into a set of periodicity features and associated source indicators by iteratively minimizing the reconstruction error. Neither layer requires any preexisting labels; they rely on the hypothesis of source-specific periodicity patterns to identify groups of spectral features and perform source separation.

In this study, the first NMF layer was used to learn 90 spectral features from the prewhitened LTSA (Fig 8):

$$\check{P}_{f,t} \approx (WH)_{f,t} = \sum_{a=1}^{90} W_{f,a} H_{a,t}. \tag{3}$$

The first NMF layer requires two constraining parameters for feature learning: the number of frames and the sparseness of spectral features. The number of frames determines the maximum amount of time-dependent information learned by each spectral feature. A short time frame may not be sufficient to capture the source-specific time-varying features; however, a

substantially long time-frame decreases the computational efficiency. Sparseness, which is a value ranging between 0 and 1, determines the approximate ratio of zero-valued elements:

$$\text{sparseness}(x) = \frac{\sqrt{n} - (\sum |x_i|)/\sqrt{\sum x_i^2}}{\sqrt{n} - 1}. \tag{4}$$

A sparseness value of 0.5 allows half of the frequency bins in each spectral feature to be activated [50]. We defined the number of frames and the sparseness as 15 min and 0.5, respectively. After feature learning and periodicity conversion, 90 periodicity vectors were factorized into four periodicity features and associated source indicators using the second NMF layer.

$$D(H)^T \approx (WH)_{p,a} = \sum_{s=1}^{4} W_{p,s} H_{s,a}, \tag{5}$$

where $D(\cdot)$ denotes the function of DFT, and $p$ represents the dimension of periodicity.

We divided the source separation into a training phase, a prediction phase, and a reconstruction phase. The training phase was to learn the source-specific spectral features from a small subset of MACHO recordings. The prediction phase was to learn the temporal activations of the source-specific spectral features from the entire dataset of MACHO recordings. After the prediction phase, each sound source was reconstructed using the source-specific spectral features and the associated temporal activations.

In the training phase, we selected the audio data of the first day of each month from November 2011 to October 2012 (34,541 audio clips) to generate a prewhitened LTSA. In the training phase, the unlabeled LTSA was processed by PC-NMF to obtain 90 spectral features and their source indicators. The source indicators were reviewed by the first author to check whether any spectral features were mislabeled by searching spectral features activated simultaneously in randomly selected audio events (e.g., shipping sounds), but recognized as different sound sources by PC-NMF. Following manual inspection, only spectral features and source indicators were saved as the prior knowledge for the prediction and reconstruction phases. After the training phase, the first author manually annotated fish choruses and cetacean vocalizations from the subsampled dataset to measure the true-positive rate and false-positive rate of the source separation model.

In the prediction phase, we only used the first NMF layer because the 90 spectral features and their source indicators were fixed. We computed a prewhitened LTSA for the audio recordings of each day and applied NMF to learn the temporal activations of the 90 spectral features. The temporal activations were initialized by random values and updated for 200 iterations.

In the reconstruction phase, we used the source-specific spectral features and newly learned temporal activations to obtain a ratio time–frequency mask. The LTSA of each sound source ($P_s$) was reconstructed by multiplying the prewhitened LTSA ($P$) and the ratio mask:

$$P_s = P \cdot \frac{W_s H_s}{WH}. \tag{6}$$

Here, we assume the mixture to be a linear sum of sources because the LTSA has been transformed to signal-to-noise ratios after the prewhitening procedure.

## Analysis of acoustic phenology

Integrated energy levels of each separated sound source were measured from the reconstructed LTSA. Because the reconstructed LTSA was linearly separated from a prewhitened LTSA, the values only approximately represented the signal strength relative to the background noise. Therefore, the integrated energy levels were normalized to a range between 0 and 1 by the

minimum and maximum values to visualize the temporal variations of each sound source. The relative intensities were applied to the measurement of the time-lagged correlations to investigate the recurring patterns. The local maximum of correlation scores within 24 h was selected to examine the diurnal pattern. We measured the correlation envelope for the lunar and seasonal cycles by using a 1-day sliding window and identified the local maximums. We also conducted the same analysis of time-lagged correlations between sound sources to examine their temporal interactions.

To model the diurnal, lunar, and seasonal changes, we applied cubic spline data interpolation [66] to fit the distribution of relative intensities. For the diurnal analysis, data within 10 min were pooled, and the 95th percentile was selected to represent each 10-min time bin. For the lunar and seasonal analyses, daily data were pooled, and the 95th percentile was used.

### Evaluation of acoustic diversity

The sound sources separated by PC-NMF were further analyzed to investigate acoustic diversity. First, we applied a threshold to the integrated energy levels to detect each separated sound source. Detected sounds separated by an interval of > 10 min were considered different acoustic events, and events lasting < 1 min were discarded. The applied criteria ensure that the identification of an acoustic event was not influenced by isolated or faint signals, thus increasing the confidence of diversity assessment. For each event, the mean spectrum was measured by pooling all the time frames.

To investigate acoustic diversity, we performed $k$-means clustering on the mean spectral levels of each separated sound source. The variation of spectral levels in each event was normalized according to its minimum and maximum values. Principal component analysis was applied to reduce the number of collinear spectral features by selecting components that explain more than 90% of the variance. Owing to the lack of information regarding the number of clusters ($k$), we allowed the algorithm to try various numbers of clusters and measured the data dispersion explained by the clustering result [43,67]. For each trial, we measured the sum of the squared Euclidean distance from each point within a cluster to the cluster centroid ($D_{i, k = n}$). Subsequently, we calculated the ratio between the mean within-cluster variation and the overall variation. The overall variation ($D_{k = 1}$) was calculated using the sum of the squared Euclidean distances from every point to the centroid of the data.

Theoretically, the ratio between the mean within-cluster variation and the overall variation should be converged when the choice of $k$ approaches the actual richness of acoustic events. However, the acoustic features of each event may vary with various factors, including behavioral context, distance to the hydrophone, and sound propagation environment. Therefore, we often chose a low $k$ value to observe the general structure of acoustic events or a high $k$ value to investigate the behavior of each sound source in detail. In this study, we quantified how well the clustering result can represent the data dispersion ($R_{k = n}$) under a given $k$ value.

$$R_{k=n} = 100 \times \left( 1 - \frac{\frac{\sum_i^n D_i}{n}}{D_{k=1}} \right). \tag{7}$$

Then, we investigated the response of $R$ with changing $k$ to determine an optimal number of clusters for each sound source (S2 Fig).

### Supporting information

**S1 Fig. Performance evaluation of audio source separation.** The performance of PC-NMF was evaluated by comparing the reconstructed LTSA of fish choruses and cetacean

vocalizations to the manual identification results. In addition, we also ran source-specific band-pass filters (according to the spectral peaks identified in Fig 2) on the prewhitened LTSA. The PC-NMF can correctly detect more fish choruses and cetacean vocalizations than the conventional band-pass filters.
(TIF)

**S2 Fig. Performance evaluation of clustering analysis.** To compare the acoustic diversity among abiotic sounds, fish choruses, and cetacean vocalizations, we iteratively changed the number of clusters and measured the percentage of variation that can be represented by cluster centroids. The curves show that cetacean vocalizations require the highest number of clusters to explain the same level of variation as fish choruses and abiotic sounds. Fish choruses require the least number of clusters to explain the same variation as abiotic sounds and cetacean vocalizations.
(TIF)

## Acknowledgments

We thank Tzu-Wei Lin and Su-Yu Chen from Central Weather Bureau, Taiwan, for their assistance on the audio archive of the MACHO.

## Author Contributions

**Conceptualization:** Tzu-Hao Lin, Yu Tsao.

**Data curation:** Tzu-Hao Lin.

**Formal analysis:** Tzu-Hao Lin.

**Funding acquisition:** Tzu-Hao Lin.

**Investigation:** Tzu-Hao Lin.

**Methodology:** Tzu-Hao Lin, Tomonari Akamatsu, Yu Tsao.

**Project administration:** Tzu-Hao Lin.

**Resources:** Yu Tsao.

**Software:** Tzu-Hao Lin.

**Supervision:** Tzu-Hao Lin.

**Validation:** Tzu-Hao Lin.

**Visualization:** Tzu-Hao Lin.

**Writing – original draft:** Tzu-Hao Lin.

**Writing – review & editing:** Tomonari Akamatsu, Yu Tsao.

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
