## [Decision Letter · Decision Letter 0]

8 Sep 2020

Dear Dr. Lin,

Thank you very much for submitting your manuscript "Sensing ecosystem dynamics via audio source separation: a case study of marine soundscapes off northeastern Taiwan" for consideration at PLOS Computational Biology.

As with all papers reviewed by the journal, your manuscript was reviewed by members of the editorial board and by several independent reviewers. In light of the reviews (below this email), we would like to invite the resubmission of a significantly-revised version that takes into account the reviewers' comments.

General assessment: The reviewers all found the paper to be an interesting application of the authors' previously-developed PC-NMF method to a larger dataset of marine soundscape data. However, in order for this paper to be a substantial contribution to the literature, there needs to be either (a) a significant computational innovation (this paper does not aim to provide that, since it relies on the authors' previously-published technical method); or (b) a clear analysis of the results as a contribution to the biological understanding of the ecosystem considered. As reviewer 3 highlights, the submitted paper does not give much information of the existing state of knowledge about the ecosystem or species interactions in the northeastern Taiwan sea.

With improvements to (b), the paper would be appropriate for PLOS CB. In your re-submission, please clearly address all referees' comments. In particular, as guest editor I highlight:

* The paper needs a statement of the current (before the study) knowledge of the northeastern Taiwan soundscape, and further discussion of the biological findings of this study.

* How exactly would someone access the data? The main website is given. Are there catalogue numbers or specific IDs/URLs that could be used to access the data?

* "two layers of non-negative matrix factorisation" is not an appropriate description, since the NMF algorithm appears to be applied only once

We cannot make any decision about publication until we have seen the revised manuscript and your response to the reviewers' comments. Your revised manuscript is also likely to be sent to reviewers for further evaluation.

Sincerely,

Dan Stowell

Guest Editor

PLOS Computational Biology

Stefano Allesina

Deputy Editor

PLOS Computational Biology

The reviewers all found the paper to be an interesting application of the authors' previously-developed PC-NMF method to a larger dataset of marine soundscape data. However, in order for this paper to be a substantial contribution to the literature, there needs to be either (a) a significant computational innovation (this paper does not aim to provide that, since it relies on the authors' previously-published technical method); or (b) a clear analysis of the results as a contribution to the biological understanding of the ecosystem considered. As reviewer 3 highlights, the submitted paper does not give much information of the existing state of knowledge about the ecosystem or species interactions in the northeastern Taiwan sea.

With improvements to (b), the paper would be appropriate for PLOS CB. In your re-submission, please clearly address all referees' comments. In particular, as guest editor I highlight:

* The paper needs a statement of the current (before the study) knowledge of the northeastern Taiwan soundscape, and further discussion of the biological findings of this study.

* How exactly would someone access the data? The main website is given. Are there cataglogue numbers or specific IDs/URLs that could be used to access the data?

* "two layers of non-negative matrix factorisation" is not an appropriate description, since the NMF algorithm appears to be applied only once

Reviewer's Responses to Questions

**Comments to the Authors:**

Reviewer #1: This paper describe a promising approach to long term unlabeled acoustic recording analysis, with a successful application and interesting results.

here are a few things I would advise to review :

There is I believe a mistake line 239 with "contains" instead of "containing" ?

Figure 2 could be displayed as the curve of the median +- the std to avoid singular peaks ?

was this figure computed over the whole recording period ?

Insights could be added on why/how the algorithm works for abiotic sounds despite the fact that there are not periodic (as seen in l.160 and fig 3)

Details on the clustering procedure, and for the rest of the method, could help for an easier read.

The method is explained, but could use more details to enhance reproductibility.

If I understood correctly, the sample size for the correct identification and false positive rates is 12 days (l.433) ?

times 24 hours times 2 LTSA per hour = 576 ?

this could be clarified with the presented results

Insights on the computation power / time needed depending on the size of the data used for learning could benefit the paper.

Reviewer #2: The review has been uploaded as an attachment.

Reviewer #3: In this manuscript the authors present a study on the northeastern Taiwan soundscape using PC-NMF, a method for source separation proposed by the authors in 2017.

They claim that “audio source separation will play a crucial role in soundscape-based ecosystem monitoring and will facilitate the assessment of acoustic diversity that is challenging in noisy environments“ and aim to provide a successful example of this in the proposed manuscript. Even though the motivation to use source separation for marine soundscapes is clear from the text, the main contribution and innovation is not.

The authors provide references to the chosen source separation method but fail to motivate the reason why source separation is needed in this case, why is such a method the best option for this study amongst the other state-of-the-art methods in audio source separation, and most importantly, they fail to expose the main findings of this study leaving the reader unsure on the contribution and innovation of this manuscript, specially taking into account that PC-NMF, its motivation and applications have already been discussed in the literature.

Overall I believe the study needs to be revisited, the methodology reconsidered and the text improved to make sure to include :

- literature review on other source separation techniques that could be used

- The motivation and justification for the chosen source separation method regarding this data

- Clear explanation and steps of the methodology used

- Justification for the technical choices made

- Statement of the current (before the study) knowledge of the northeastern Taiwan soundscape

- Clear description of the findings of their study and their impact

Here are some additional comments the authors might find useful to improve their study and text :

Regarding source separation :

From the text, the authors seem to have some confusion over the terms “simultaneously” and “overlapping”, which unfortunately does not help motivate the need for advanced source separation techniques in this study. An example of this confusion can be found at line 109 where the authors wrongly state that “signals recorded simultaneously cannot be separated on the basis of clustering” as one can imagine a mixture of a low and a high pitched source to be separated through “clustering” as long as their overlap remains in time and not frequency. Other examples can be found at line 110 and 120.

If two sounds happen simultaneously they overlap in time but it does not mean that they also overlap in frequency. As a matter of fact, in source separation we often use time-frequency representations to exploit this : less overlap between sources in the frequency domain allows for simpler techniques to separate sources. Of course if the sources do not overlap in time, then the separation can simply be done in the time domain.

In addition, the different sound properties of sources also play a key role when deciding which source separation method to use. For example stationary sources can easily be separated and removed through Wiener filtering. Therefore the use of NMF to remove electrical noise ( visibly stationary in Figure 1 and 2) is unjustified.

As a summary, here are some questions to take into consideration (and discuss) when deciding which method to use in source separation:

- Do sources overlap in time?

—> No : simple separation in time domain

-> Yes : Do they overlap in frequency?

-> No : simple separation in time-frequency domain

-> Yes : Do the sources share properties (e.g. stationary, impulsive, broadband, harmonic … )?

-> No : basic modelling and filtering

-> Yes : advanced source separation techniques such as NMF

The recordings in this study can be divided into four sources : abiotic sound, fish chorus, cetacean vocalisations and electrical noise. As the electrical noise is stationary it should be removed through filtering in an initial pre-processing stage, leaving abiotic, fish and cetaceans as sources to be considered further. From these three sources, we could safely assume the abiotic sounds to have considerably distinct properties to the biological sources, making the separation between fish and cetaceans the real technical challenge in this study. However, from the text we find out that these two types of biological sounds do not always overlap in time (e.g. Line 179) and furthermore, they seem to only overlap in frequency between 1 and 3kHz (line 135 onwards), raising questions on the real need for advanced source separation methods such as PC-NMF. The authors did not provide any discussion on this, nor strong arguments for the use of PC-NMF in this study.

- Line 51 : “This study applied machine learning techniques to analyze underwater recordings transmitted from a deep seafloor observatory.” : what are the other machine learning techniques used in this study? Be aware NMF is not always considered machine learning in the literature, specially on its basic form like in PC-NMF.

- Line 139 : “Despite the significant frequency overlap among sound sources… ” —> 2kHz overlap is generally not considered significant, therefore if it is significant in this case please explain why.

- Line 190 : “By applying a threshold of 0.05.. “ : Provide justification.  

Regarding PC-NMF :

My understanding of PC-NMF (the text was not clear on this so I went and read reference 33) : First apply standard NMF approximating a time-frequency representation of the sound recording into a multiplication of two non-negative matrices, one known as the “dictionary” containing the spectral basis and another known as the activation matrix telling us when and how those spectral basis are present in the signal. Now the goal becomes to determine which spectral basis belong to the target source we want to separate, in the case of underwater recordings, the biological sounds. In PC-NMF they assume the biological sounds to be periodic unlike the interfering anthropogenic/environmental sounds, and so they analyse the repetitions of patterns in the activation matrix by taking the FFT of every row in the activation matrix. Every spectral template (i.e. row in activation matrix) that peaks at the same time in the FFT is considered to be originated from the same biological source. In this way one can group spectral basis into different biological sources and noise to continue with the standard separation in NMF.

Therefore the interesting questions about PC-NMF are :

-> Does the main assumption of biological sound periodicity always hold? One can imagine boats also producing period sounds, how does this affect PC-NMF? How important is the difference in period between sound sources?

-> How many spectral basis (i.e. rank) should one pick to make sure there is no confusion between sources?

—> What is the a priori knowledge needed to be able to allocate spectral basis to certain species (i.e. sound sources)? Knowing how to cluster spectral templates does not imply knowing its sound source.

None of the above were discussed in the presented text.

- Abstract : “two layers of non-negative matrix factorisation” : PC-NMF seems to perform only one NMF optimisation, and therefore has only “one layer”. A second layer of NMF could for example be taking the learnt templates of a source and fixing them on another semi-supervised optimisation round —> does this happen here?

- Line 52 : Which training?

- Line 130 onwards : How did you decomposed the LTSA into four independed sound sources? What a priori knowledge did you use? This should be discussed in depth.

-Line 159 : “Compared to the pattern observed in abiotic sounds, the patterns of the two biophonic sound sources showed clear periodic structures.” — > Where can we see these clear periodic structures?

- Line 192 onwards : How did the clustering work? What a priori information was used? There should be a major focus on this as it is where the contribution of this paper could rely.

- Line 241 : “… because most of the biological activities occur periodically. “ —> Please provide justification or reference or this claim.

- Paragraph starting at Line 246 : How was the process manually modified to increase its performance? Which were the spectral basis incorrectly assign? And most importantly, why?

- Line 333 : I’m confused, which training data do you refer to here?

- Line 416 : Justify the choice of rank 90 or at least provide a discussion on the rank choice impact.

- Paragraph starting at line 464 is not clear, more detail on how to perform the clustering is key for other researchers trying this method to find sources.

 

Regarding the evaluation, contributions and innovation :

Possible contributions :

- Abstract : Identification of diurnal and seasonal interactions between cetaceans and soniferous fish

- Line 55 : Finding : continental shelf environment has a high diversity of sound-producing animals, and the community structure changes with daily, lunar, and seasonal cycles.

- Line 58: “A database of anthropogenic noise has also been established and used to evaluate how human activities may affect the behavior of sound-producing animals.” —> Did not find any other reference to this in the text.

In order to better understand if these are contributions background is needed. What did we know before this study? It is unclear what information is known and what is found through the presented analysis, and its impact. How do these findings advance the field?

- Line 156 : “We noticed that audio data were heavily corrupted … Therefore, these data were not included in the subsequent analyses.” —> Please provide further explanation for this as unjustified partial use of data is not scientifically acceptable.

- Line 214 onwards : The cetacean clustering and analysis should de discussed in depth alongside some data representation supporting the claims such as the one on Line 218.

- Line 269: Which type of fish? How did you identified it? Where is the report on their spectral, diurnal and seasonal patterns?

- Line 294 : “According to previous experiments, … “ —> which ones?

- Paragraph starting at 432 : Does this mean that the results presented were issued from a supervised approach? If manually labelled data was used to ‘learn’ spectral basis of the sources through an initial NMF and were then fixed for a semi-supervised NMF round in the rest of the data, please make sure to state earlier in the text section that the results are not issued from an unsupervised method as is currently claimed. Example: how does the statement at line 122 “First, we applied audio visualization and source separation techniques to demonstrate the feasibility of separating biotic and abiotic sounds in an unsupervised manner” with the statement at line 439 “The analysis of the entire dataset was conducted in a supervised manner” ?

- Line 461 : Justification missing for the 10 and 1 min choices.

Reviewer #4: In general it is a well-written article.

Methodologically, I would suggest that you be more detailed about the technique used, especially if you want to attract reader engineers who want to contribute to the algorithms.

**Have all data underlying the figures and results presented in the manuscript been provided?**

Reviewer #1: Yes

Reviewer #2: **No: **The authors provided an URL to access the audio recordings. However, the GDMS website requires subscription and is for Taiwan users only.

Reviewer #3: Yes

Reviewer #4: Yes

PLOS authors have the option to publish the peer review history of their article (what does this mean?). If published, this will include your full peer review and any attached files.

Reviewer #1: **Yes: **PAUL BEST

Reviewer #2: No

Reviewer #3: No

Reviewer #4: No
---

## [Decision Letter · Decision Letter 1]

16 Dec 2020

Dear Dr. Lin,

Thank you very much for submitting your manuscript "Sensing ecosystem dynamics via audio source separation: a case study of marine soundscapes off northeastern Taiwan" for consideration at PLOS Computational Biology. As with all papers reviewed by the journal, your manuscript was reviewed by members of the editorial board and by several independent reviewers. The reviewers appreciated the attention to an important topic. Based on the reviews, we are likely to accept this manuscript for publication, providing that you modify the manuscript according to the review recommendations.

Thank you for your work on revising the manuscript. Three reviewers are happy with the paper in its current form. Reviewer 3 raises various issues, including various points in which the article text is not sufficiently precise about technical terms. I agree with Reviewer 3 on this. However, unlike Reviewer 3, I consider this paper to be a good fit to the journal PLOS Comp Biol, since our scope concerns application and not just the theoretical innovation of computational methods.

In your resubmission, please make sure you address Reviewer 3's issues, including the issues of technical precision. For example:

* The abstract should not claim "A source separation model ... was developed" since the source separation model is one developed in prior work.

* "minimal supervision" is not the same as "unsupervised". Authors must be careful, and not claim a method is "unsupervised" when the procedure involves manual interventions to achieve the desired per-class outputs. It can be correct to label a component, for example a single pass of PC-NMF, as an unsupervised process. However, the current wording in the abstract is imprecise, and could mislead a reader into believing the overall procedure is unsupervised.

* The definition of "A BSS model aims..." needs to be made technically correct.

Please see the reviews for further comments.

Sincerely,

Dan Stowell

Guest Editor

PLOS Computational Biology

Stefano Allesina

Deputy Editor

PLOS Computational Biology

[LINK]

Thank you for your work on revising the manuscript. Three reviewers are happy with the paper in its current form. Reviewer 3 raises various issues, including various points in which the article text is not sufficiently precise about technical terms. I agree with Reviewer 3 on this. However, unlike Reviewer 3, I consider this paper to be a good fit to the journal PLOS Comp Biol, since our scope concerns application and not just the theoretical innovation of computational methods.

In your resubmission, please make sure you address Reviewer 3's issues, including the issues of technical precision. For example:

* The abstract should not claim "A source separation model ... was developed" since the source separation model is one developed in prior work.

* "minimal supervision" is not the same as "unsupervised". Authors must be careful, and not claim a method is "unsupervised" when the procedure involves manual interventions to achieve the desired per-class outputs. It can be correct to label a component, for example a single pass of PC-NMF, as an unsupervised process. However, the current wording in the abstract is imprecise, and could mislead a reader into believing the overall procedure is unsupervised.

* The definition of "A BSS model aims..." needs to be made technically correct.

Please see the reviews for further comments.

Reviewer's Responses to Questions

**Comments to the Authors:**

Reviewer #1: Some insights were added about the learnt / discovered features of the studied soundscape. The paper thus contributes to the general knowledge of the ecosystem considered.

Reviewer #2: Review uploaded as an attachment.

Reviewer #3: The authors addressed some of the concerns mentioned in the first review but the remaining lack of rigour and unclear technical innovation make this manuscript unsuitable for publication in the PLOS Computational Biology journal where I understand methodologies to be the driving force.

Please find some examples of the lack of rigour:

* line 119 : the definition of BSS is wrong: "A BSS model aims to learn the temporal variations of a mixture using a set of unknown sound sources."

BSS is employed when there is no information about the individual sources (e.g. placement, acoustic conditions, number ...) and only the joint effect (i.e. mixture) is known.

* line 493: "LTSA" example of one of the multiple acronyms used without definition

* The authors claim the method used here is unsupervised (e.g. line 135 or line 190) yet further down in the text we read otherwise. The "training phase to learn the source-specific spectral features" (l.544) renders the method to be at the very least semi-supervised as these spectral basis are then fixed (l.564). Further if one takes into account in that the author "checked" (l.555) whether any spectral features were mislabeled or that the "author manually annotated" (l.560) it becomes apparent that the method presented is clearly not unsupervised.

* Equation (6) assumes the mixture to be a linear sum of sources - is this the case? Please explain why this assumption is valid in this case.

Regarding the lack of innovation: I have not found an addition or extension to the PC-NMF method presented in this text and following these comments from the authors in response to my previous review

"However, parameter tuning is labor-intensive and often needs to repeat when analyzing data collected from a new environment."

"The application of PC-NMF can greatly reduce the labor-intensive nature of the works and the need of prior knowledge.."

I am under the impression the main contribution of the presented study is the application of an existing method to new data, which I do not regard as technically innovative.

I would also like to point out the organisation and clarity of presentation also require significant improvement.

I cannot not comment on the biological findings as I am not qualified to do so.

Closing remarks on some of the responses to my previous review:

"The third reviewer may confuse the application of PC-NMF in different stages of data processing."

 yes, I am indeed confused about the method in this paper yet I am very acquainted with NMF and its variants.

"We also added more paragraphs to demonstrate how applied techniques can improve acoustic diversity assessment and reveal the difference in spectral characteristics and phenological patterns between groups of marine soniferous animals "

 What are (the) "applied techniques"?

“Low audio quality owing to the simultaneous recording of multiple sound sources increases clustering uncertainty.”

 I do not understand this. Simultaneous recording of multiple sound sources means that you recorded different sound sources at the same time (could mean with multiple mics). I think you mean "the recording of simultaneous sound sources". In any case, how come the sound sources being or not simultaneous affect in any way the audio quality?

"The dataset used in this study contains multiple types of biotic and abiotic sound sources that may be overlapping in time or frequency domains (Fig 2 & 3 in the revised manuscript), and each sound source also possesses spectral variability. We demonstrated that the applied technique of audio source separation not only reduces the interference among sound sources but also enables further applications, such as the variability assessment of source behaviors (Fig 4 & 7 in the revised manuscript). Most importantly, the applied technique can transform single-channel audio into multiple channels that encode the dynamics of biophony, geophony, and anthropophony. The outcomes will expand the deliverables of soundscape monitoring that are difficult to achieve by using conventional approaches."

 This should be in the abstract and/or introduction as it is very clearly explained.

"The electrical noise in the MACHO recordings is not stationary. "

 Based on the spectrograms of Figure 2, the electrical noise is certainly slow varying compared to other sources and quite obviously stationary at least four frequency bins (around 25, 30, 45 and 55kHz) from what I can see. I believe simple de-noising techniques based on Wiener filtering would remove the electrical noise if applied at a lower time scale windows (e.g. 15min instead of 11 hours).

Reviewer #4: I keep my first review. To Accept

**Have all data underlying the figures and results presented in the manuscript been provided?**

Reviewer #1: Yes

Reviewer #2: Yes

Reviewer #3: Yes

Reviewer #4: Yes

PLOS authors have the option to publish the peer review history of their article (what does this mean?). If published, this will include your full peer review and any attached files.

Reviewer #1: No

Reviewer #2: No

Reviewer #3: No

Reviewer #4: No
---

## [Editor Report · Decision Letter 2]

12 Jan 2021

Dear Dr. Lin,

We are pleased to inform you that your manuscript 'Sensing ecosystem dynamics via audio source separation: a case study of marine soundscapes off northeastern Taiwan' has been provisionally accepted for publication in PLOS Computational Biology.

Thank you for the detailed response, for the "diff" document, and for the many amendments to the paper to account for the review feedback. I now judge that the imprecisions and misstatements raised in review have been addressed. The paper is much improved from the first submission, and appropriate for publication.

Best regards,

Dan Stowell

Guest Editor

PLOS Computational Biology

Stefano Allesina

Deputy Editor

PLOS Computational Biology

---

## [Editor Report · Acceptance letter]

2 Feb 2021

PCOMPBIOL-D-20-01169R2 

Sensing ecosystem dynamics via audio source separation: a case study of marine soundscapes off northeastern Taiwan

Dear Dr Lin,

I am pleased to inform you that your manuscript has been formally accepted for publication in PLOS Computational Biology. Your manuscript is now with our production department and you will be notified of the publication date in due course.

With kind regards,

Alice Ellingham
